# Immune Cell Infiltration in the Microenvironment of Liver Oligometastasis from Colorectal Cancer: Intratumoural CD8/CD3 Ratio Is a Valuable Prognostic Index for Patients Undergoing Liver Metastasectomy

**DOI:** 10.3390/cancers11121922

**Published:** 2019-12-02

**Authors:** Jianhong Peng, Yongchun Wang, Rongxin Zhang, Yuxiang Deng, Binyi Xiao, Qingjian Ou, Qiaoqi Sui, Jing Xu, Jiayi Qin, Junzhong Lin, Zhizhong Pan

**Affiliations:** 1Department of Colorectal Surgery, Sun Yat-sen University Cancer Center, State Key Laboratory of Oncology in South China, Collaborative Innovation Center for Cancer Medicine Guangzhou, Guangzhou 510060, China; pengjh@sysucc.org.cn (J.P.); zhangrx@sysucc.org.cn (R.Z.); dengyx@sysucc.org.cn (Y.D.); xiaoby@sysucc.org.cn (B.X.); ouqj@sysucc.org.cn (Q.O.); suiqq@sysucc.org.cn (Q.S.); qinjy@sysucc.org.cn (J.Q.); 2Department of Experimental Research, Sun Yat-sen University Cancer Center, State Key Laboratory of Oncology in South China, Collaborative Innovation Center for Cancer Medicine Guangzhou, Guangzhou 510060, China; wangych@sysucc.org.cn (Y.W.); xujing@sysucc.org.cn (J.X.)

**Keywords:** colorectal cancer, liver oligometastasis, microenvironment, immune cell, CD8/CD3 ratio

## Abstract

Background: A comprehensive investigation into immune cell infiltration provides more accurate and reliable prognostic information for patients with colorectal liver oligometastases (CLO) after liver metastasectomy. Methods: Simultaneous detection of the immune constituents CD3^+^, CD8^+^, Foxp3^+^ T, and α-SMA^+^ cells in the liver oligometastasis of 133 patients was conducted using a four-colour immunohistochemical multiplex technique. Immune cells were quantified, and tumour-infiltrating lymphocyte (TIL) ratios were subsequently calculated. Correlation analysis was performed using Pearson’s correlation. Recurrence-free survival (RFS) and overall survival (OS) for TIL ratios were analysed using the Kaplan–Meier method and Cox regression models. Results: Significantly fewer CD3^+^, CD8^+^, and Foxp3^+^ T cells were observed in the intratumoural region than in the peritumoural region of liver metastases. CD3^+^, CD8^+^, Foxp3^+^ T, and α-SMA^+^ cells showed significantly positive correlations with each other both in the intratumoural and peritumoural regions of liver metastases. Only the CD8/CD3 TIL ratio demonstrated a positive correlation between intratumoural and peritumoural regions of liver metastases (r = 0.541, *p* < 0.001). Patients with high intratumoural CD8/CD3 ratios had significantly longer 3-year RFS (59.0% vs. 47.4%, *p* = 0.035) and 3-year OS rates (83.3% vs. 65.8%, *p* = 0.007) than those with low intratumoural CD8/CD3 ratios. Multivariate analyses revealed that the intratumoural CD8/CD3 ratio was independently associated with RFS (HR = 0.593; 95% CI = 0.357–0.985; *p* = 0.043) and OS (HR = 0.391; 95% CI = 0.193–0.794; *p* = 0.009). Conclusion: These findings offer a better understanding of the prognostic value of immune cell infiltration on liver oligometastasis from colorectal cancer.

## 1. Introduction

Distant metastasis remains a leading cause of treatment failure and disease-related death for colorectal cancer patients [1]. Liver metastasis is the most frequent pattern of distant metastases and might not be considered a unified disease entity due to the differences in the metastatic burden of different cancer types and various prognoses [2,3]. Recently, oligometastatic disease was highlighted as a disease state that exists in a transitional zone between localized and widespread systemic diseases, indicating a limited growth potential [3,4]. Although patients with colorectal liver oligometastasis (CLO) achieved a 45.9% overall survival (OS) rate after liver resection within five years, we previously found that 57.3% of the patients developed postoperative recurrence, and 16.0% developed early recurrence [5]. Therefore, the management of CLO is challenging, and screening out specific prognostic risk subgroups is urgently needed to provide a more personalized postoperative treatment approach [6,7].

Growing evidence indicates that the host immune response against CRC has a crucial effect on tumour progression and survival [8,9]. In fact, the inflammatory and immunological responses observed in CRC are heterogeneous among patients. The immune microenvironment, which comprises immune cells infiltrating or surrounding the tumour, is the appropriate milieu for mounting an immune response against CRC [10]. Tumour-infiltrating lymphocytes (TILs) are a dominant immune component found in the stroma of primary tumour CRC samples. High densities of CD3^+^, CD8^+^, and Foxp3^+^ T cells were associated with prolonged survival and a significant therapeutic response in CRC patients [11,12,13,14]. Furthermore, TIL localization might also be associated with their function and clinical effect. Turksma et al. revealed that CD8^+^ lymphocytes located in the tumour stroma did not possess prognostic significance, whereas CD8^+^ lymphocytes localized in the tumour epithelium positively affected postoperative recurrence [15]. Cancer-associated fibroblasts (CAFs) are another abundant cell type in the tumour immune microenvironment and can be identified by markers such as α-smooth muscle actin (α-SMA) [16]. CAFs are widely involved in tumour proliferation, invasion, and metastasis in various malignancies. More importantly, CAFs interact with TILs and drive changes in the functional status of T cells [17]. Therefore, the presence of CAFs in the tumour microenvironment deserves further investigation.

To the best of our knowledge, the local immune response impacts the clinical course not only from the primary tumour but also from metastatic liver lesions [18,19]. Thus, examining the effect of the tumour microenvironment on liver metastases might provide deeper insight into the appropriate immune biomarker ratio that may provide more accurate and reliable prognostic information for CLO. The present study applied a novel tyramide signal amplification (TSA) multiplexing technique to enable the simultaneous examination of four distinct markers, including CD3, CD8, Foxp3, and α-SMA. Accordingly, we aimed to (1) describe the tumour microenvironment of oligometastatic liver lesions in both intratumoural and peritumoural regions and (2) evaluate the prognostic value of intratumoural and peritumoural TIL ratios in liver oligometastases from patients with CRC who underwent liver metastasectomy.

## 2. Materials and Methods

### 2.1. Patient Selection

We recruited consecutive patients with CLO who had undergone liver metastasectomy from June 1999 to December 2016 at Sun Yat-sen University Cancer Center, Guangzhou, China. The eligibility criteria were as follows: (1) pathologically confirmed colorectal adenocarcinoma; (2) radiologically diagnosed as a single colorectal liver metastasis; (3) underwent potentially curative resection for both colorectal primary tumour and liver metastasis; and (4) adequate metastasis specimens were available for analysis. The exclusion criteria were as follows: (1) the presence of preoperative extrahepatic metastases; and (2) a history of prior liver resection. Demographic and clinicopathological characteristics were retrieved from the electronic medical record system, and follow-up data were collected from the follow-up system. Recurrence risk in patients after liver resection was evaluated according to the Memorial Sloan–Kettering Cancer Center clinical risk score (MSKCC-CRS) [20]. The treatment strategy and operability of liver metastases were determined for each patient based on consensus of the multidisciplinary team (MDT).

The current study was approved by the Institutional Research Ethics Committee of Sun Yat-sen University Cancer Center (approval number: GZR2017-006) and was conducted in accordance with the principles outlined in the Declaration of Helsinki of the World Medical Association. Informed consent for the use of tissue samples and medical information for clinical research was obtained from the patients before surgery.

### 2.2. Four-Colour Immunohistochemical Multiplex

Paraffin-embedded tissues were sectioned into 4-μm-thick sections. The tissue sections were dewaxed in xylene, rehydrated, and rinsed in graded ethanol solutions. Prior to antigen retrieval in heated citric acid buffer (pH 6.0), 0.3% H_2_O_2_ was applied to reduce the activity of endogenous peroxidase for 10 min at room temperature. Information on of the primary antibodies and the multiplex staining reagents is provided in Appendix A. Four different primary antibodies (CD3, CD8, Foxp3, and α-SMA) were applied through sequential rounds of staining, each including a protein block with 3% bovine serum albumin (BSA), a secondary horseradish peroxidase-conjugated polymer (EnVision Detection Systems, Dako Cytomation, Carpinteria, CA, USA), and a different fluorophore using TSA. The TSA covalent reaction was stopped by additional antigen retrieval in heated citric acid buffer (pH 6.0) for 25 min to remove the bound primary antibody before the next step in the sequence. Nuclei were counterstained with DAPI (4,6-diamidino-2-phenylindole, Thermo Scientific, Rockford, IL, USA) after all four human antigens were labelled. Quality control was also conducted using standard immunohistochemistry (IHC).

### 2.3. Multispectral Imaging

The stained slides were scanned using the Vectra Multispectral Imaging System version 2 (PerkinElmer, Waltham, MA, USA), with one raw image comprising five stitched 200× multispectral image cubes for intratumoural or peritumoural tissue regions. Each 200× multispectral image cube captures the fluorescent spectra at 20 nm wavelength intervals from 420 to 720 nm with identical exposure times. The filter cubes used for multispectral imaging were DAPI (440–680 nm), CD3 (520 nm), CD8 (690 nm), Foxp3 (620 nm), α-SMA (570 nm), and pan-keratin (520 nm) (Appendix A).

### 2.4. Spectral Unmixing and Phenotyping

To separate each multispectral image cube into its individual components (spectral unmixing) and enable the colour-based identification of T-cell subtypes and α-SMA-positive cells, the Nuance Image Analysis software (Perkin Elmer, Waltham, MA, USA) was used to create a spectral library containing the emitting spectral peaks of all fluorophores obtained from single stained slides for each marker and associated fluorophore. All spectrally unmixed and segmented images were then analysed using inForm 2.1 image analysis software. Based on the identification of the DAPI-stained nuclear/cell morphological features and their patterns of fluorophore expression, immune and other cells were phenotyped into five classes: CD3^+^CD8^+^ T cells, CD3^+^Foxp3^+^ T cells, CD3^+^CD8^−^T cells, α-SMA^+^ cells, and other cells.

### 2.5. Quantification of T-Cells and a-SMA-Positive Cells

The total cells of each image were adjusted to tissue areas. The numbers of each T cell subtype and α-SMA-positive cells in each image were calculated based on the identification of the DAPI-stained nuclei. The quantification of single-positive or double-positive cells was counted as follows: Number of total cells × positivity. An average of at least five representative fields at 200× magnification of each area was selected and expressed as the number of cells per field.

### 2.6. Follow-Up

The patients were monitored through subsequent visits every 3 months for the first 2 years and then semi-annually for the next 5 years after liver metastasectomy. Specific evaluations, including computerized tomography (CT) imaging of the chest, abdomen, and pelvis, were conducted at 3, 6, 12, and 18 months, then at 2 years, and annually thereafter. A colonoscopy was conducted every year. Liver magnetic resonance imaging (MRI) was used to confirm suspicious lesions indicated on CT or in patients with increased carcinoembryonic antigen (CEA) or carbohydrate antigen 19-9 (CA19-9) levels but negative CT results. OS was defined as the interval from the date of liver metastasectomy to the date of death from any cause or to the last follow-up. Recurrence-free survival (RFS) was defined as the interval from the date of liver metastasectomy to the date of disease recurrence, death, or the last follow-up. Random censoring was applied to patients without recurrence or death at the last follow-up date. The final follow-up visit occurred in January 2018.

### 2.7. Statistical Analysis

Statistical analyses were performed using SPSS 24.0 software (IBM, Chicago, IL, USA) and GraphPad Prism 7 software (GraphPad Software, Inc, San Diego, CA, USA). Categorical variables were presented as percentages and compared using the chi-square (χ^2^) test or Fisher’s exact test. Continuous variables were presented as the means with 95% confidence intervals (CI) and compared using Student’s t-tests. The correlations between two different immune cell quantification or ratios in intratumoural and peritumoural regions of liver metastases were assessed using Pearson’s correlation coefficient (r). The cut-off value of the calculated ratio was determined by the median and then classified into either a high or low ratio group. The Kaplan–Meier method was used to estimate the survival rates for the different groups, and differences in survival were compared using the log-rank test. Variables for which *p* < 0.10 in the univariate Cox models were further assessed in multivariate Cox proportional models. A *p* < 0.05 was considered statistically significant.

## 3. Results

### 3.1. Patient Characteristics

A total of 133 eligible patients with curative resection of liver oligometastases were retrospectively analysed in this study. The clinical characteristics are summarized in Table 1. All patients were followed up for a median of 34.5 months (range: 2.0–143.3 months) after liver metastasectomy. Up to January 2018, 63 (47.4%) patients experienced tumour recurrence, including 33/72 (45.8%) patients with tumour recurrence, and 38 (28.6%) patients died of tumour progression. The 3- and 5-year RFS rates for the total patients investigated were 53.7% and 46.1%, respectively, and the 3- and 5-year OS rates were 74.7% and 65.2%, respectively.

### 3.2. The Number and Ratio of TILs in the Intratumoural and Peritumoural Regions of Liver Metastases

Pan-keratin expression was used to identify specific regions of liver oligometastases from colorectal cancer (Figure 1). Pan-keratin was found to be highly expressed in cytoplasm of the tumour cells but not in normal hepatocytes. Appendix A shows the good immunofluorescence quality control performed in the present study. Representative images of all analysed cellular phenotypes are shown in Figure 1. The CD3^+^, CD8^+^, Foxp3^+^ T, and α-SMA^+^ cells were enriched in the peritumoural regions of liver metastases (Figure 2a) and were present in the intratumoural regions of liver metastases (Figure 2b). The mean values of the absolute numbers for each T-cell subtype and α-SMA-positive cells in the peritumoural regions of liver metastases were as follows: CD3 (90.5; 95% CI = 66.9–114.2), CD8 (43.7; 95% CI = 28.4–59.0), Foxp3 (3.8; 95% CI = 2.6–5.0), and α-SMA (212.3; 95% CI = 179.0–245.6). The mean values of the absolute numbers of the immune cells in the intratumoural region of the liver metastasis were as follows: CD3 (482.4; 95% CI = 425.4–539.5), CD8 (168.5; 95% CI = 183.2–198.8), Foxp3 (7.0; 95% CI = 5.1–8.9), and α-SMA (221.0; 95% CI = 179.5–262.5). The number of CD3^+^, CD8^+^, and Foxp3^+^ T cells was significantly lower in the intratumoural region than in the peritumoural region (all *p* < 0.01, Figure 3a–c). However, the number of α-SMA^+^ cells was comparable between the two regions (*p* = 0.874, Figure 3d).

The mean CD8/CD3 ratio was not significantly different between intratumoural and peritumoural regions of liver metastases (0.32 vs. 0.33, *p* = 0.841, Figure 3e). The mean Foxp3/CD3 ratio was significantly higher in the intratumoural region than in the peritumoural region (0.11 vs. 0.01, *p* < 0.001, Figure 3f), while the mean CD3/α-SMA ratio was significantly lower in the intratumoural region than in the peritumoural region (1.92 vs. 10.82, *p* < 0.001, Figure 3g). The median values of each calculated TIL ratio in the intratumoural regions of liver metastases were as follows: CD8/CD3 ratio (0.24), Foxp3/CD3 ratio (0.03), and CD3/α-SMA ratio (0.27). The median values of TIL ratios in the peritumoural regions of liver metastases were as follows: CD8/CD3 ratio (0.31), Foxp3/CD3 ratio (0.008), and CD3/α-SMA ratio (2.585). Representative images of high and low CD8/CD3 ratios in peritumoural and intratumoural regions of liver metastases are presented in Figure 4.

### 3.3. Correlation Analysis of TILs and α-SMA-Positive Cells

The Pearson correlation coefficients (r) of each TIL and α-SMA^+^ cell number are shown in Figure 5a. The results of our statistical analyses demonstrated that the numbers of both TIL and α-SMA^+^ cells showed a significantly positive correlation with each other both in the intratumoural and peritumoural regions of liver metastases. Moreover, the density of CD8^+^ T cells was strongly associated with CD3^+^ T cells in the peritumoural region (r = 0.701, *p* < 0.001, Figure 5a). In addition, the number of CD3^+^ T cells in the peritumoural region was only weakly associated with the number of CD3^+^ and CD8^+^ T cells in the intratumoural region (r = 0.318 and r = 0.277, both *p* < 0.001, Figure 5a).

Regarding the correlation of each TIL ratio between the two regions of liver metastases, only the CD8/CD3 ratio in the intratumoural region was positively correlated with the CD8/CD3 ratio in the peritumoural region (r = 0.541, *p* < 0.001, Figure 5b). The Foxp3/CD3 and CD3/α-SMA ratios did not show significant correlations between the two regions metastases (r = 0.194 and r = 0.132, both *p* > 0.05, Figure 5c–d).

### 3.4. Relationship Between TIL Ratios and Clinicopathological Features

The associations between clinicopathological features and each TIL ratio in the two regions of liver metastases are summarized in Table 2. Liver metastasis from colon cancer was associated with a higher intratumoural CD8/CD3 ratio (73.5% vs. 52.3%, *p* = 0.011), peritumoural CD8/CD3 ratio (72.6% vs. 54.9%, *p* = 0.035), and peritumoural Foxp3/CD3 ratio (77.8% vs. 55.8%, *p* = 0.013) than rectal cancer. A low peritumoural CD8/CD3 ratio was more likely to be found the during lymph node (N)-positive stage (76.1% vs. 56.5%, *p* = 0.017). No significant association was found between the TIL ratio and other clinicopathological characteristics.

### 3.5. Association between TIL Ratios and Survival

The survival curves showed that the 3-year RFS rate was significantly higher in the high intratumoural CD8/CD3 ratio group than in the low intratumoural CD8/CD3 ratio group (59.0% vs. 47.4%, *p* = 0.035, Figure 6a). Similarly, the high intratumoural CD8/CD3 ratio group also exhibited a higher 3-year OS rate than did the low intratumoural CD8/CD3 ratio group (83.3% vs. 65.8%, *p* = 0.007, Figure 6b). Either a high or low intratumoural Foxp3/CD3 ratio and CD3/α-SMA ratio presented with comparable 3-year RFS and OS rates (all *p* > 0.05, Figure 6c–f). Additionally, peritumoural CD8/CD3, Foxp3/CD3, and CD3/α-SMA ratios were not significantly associated with RFS and OS (all *p* > 0.05, Figure 7). Additionally, we analysed the association between TIL number and survival but did not find any additional value over the ratios (Appendix A).

As shown in Table 3, univariate analyses revealed that high intratumoural CD8/CD3 ratios (HR = 0.585; 95% CI = 0.353–0.968; *p* = 0.037) were significantly positive predictors of 3-year RFS, while N stage 1–2 (HR = 1.844; 95% CI = 1.044–3.256; *p* = 0.035) was a significant negative predictor of 3-year RFS. Multivariate analysis showed that the intratumoural CD8/CD3 ratio (HR = 0.593; 95% CI = 0.357–0.985; *p* = 0.043) and N stage (HR = 1.806; 95% CI = 1.018–3.202; *p* = 0.043) were independent predictors of 3-year RFS. Regarding 3-year OS, univariate analysis (HR = 0.392; 95% CI = 0.193–0.793; *p* = 0.009) and multivariate analysis (HR = 0.391; 95% CI = 0.193–0.794; *p* = 0.009) showed that only a high intratumoural CD8/CD3 ratio was a significant positive predictor.

## 4. Discussion

The precise understanding of the immune response in CLO requires a comprehensive description of its microenvironmental complexities. In the present study, we first applied a four-colour immunohistochemical multiplex system to evaluate immune cell infiltration in liver oligometastases of CRC. Here, subpopulations of TIL cells and α-SMA-positive cells were identified both in the intratumoural and peritumoural regions of liver metastases. In addition, we found that a high intratumoural CD8/CD3 ratio was a strong prognostic factor for prolonged postoperative survival independent of the liver tumour size, timing of metastasis, preoperative CEA level, and primary lymph node metastasis. These results indicate that the use of multiple markers is important, as it can allow simultaneous calculation of intratumoural CD8/CD3 ratios for predicting prognosis.

Previous studies have revealed that intratumoural CD3^+^ and CD8^+^ T cell counts in liver metastases were significant predictors of survival and recurrence for patients with colorectal liver metastases (CRLM) who had undergone liver metastasectomy. The randomized EORTC study 40983 demonstrated that high CD3^+^ lymphocyte counts at the liver tumour front could serve as positive prognostic factors for CRLM patients [21]. Katz et al. found that a high number of CD8^+^ T cells in the liver tumour independently correlated with 10-year survival following CRLM resection [22]. Similarly, Nazemalhosseini-Mojarad et al. suggested that a high level of tumour-infiltrating CD8^+^ T lymphocytes was an independent prognostic factor for favourable patient survival [12]. Recently, Wang et al. reported that a high immunoscore consisting of the densities of CD3^+^ and CD8^+^ T cells in the intratumoural and peritumoural regions had greater power to significantly predict longer RFS and OS compared to the conventional clinical risk score (CRS) system [23]. These findings suggest a role for CD3^+^ T cells in activating an immune response in CRLM, especially cytotoxic CD8^+^ T cells, which could act as direct mediators of tumour killing and enhanced activation status of TIL. However, our data revealed that the number of CD8^+^ T cells showed a positive correlation with CD3^+^ T cells, especially in peritumoural metastases (r = 0.701, *p* < 0.001), which indicates that the balance between CD8^+^ and CD3^+^ T cells may be an authentic determinant of a patient’s capacity to mount an effective immune response against intrahepatic metastases.

Because the balance between CD8^+^ and CD3^+^ T cells was very important to reflect the immune response in the liver metastasis microenvironment, TIL ratios might be a better predictor of prognostic outcome than individual T cell counts. Katz et al. found that the CD8/CD3, Foxp3/CD4, and Foxp3/CD8 ratios were independent predictors of prognostic outcome, whereas the individual CD8^+^ and Foxp3^+^ T cell counts were not [24]. In addition, a recent study reported that individual densities of intratumoural or peritumoural CD8^+^ and Foxp3^+^ cells were not prognostic of survival, while the intratumoural CD8^+^/Foxp3^+^ ratio was an independent predictor of survival (HR = 0.43, 95% CI = 0.19–0.95, *p* = 0.032) [25]. We hypothesize that several reasons might account for these observed findings: (1) the CD8/CD3 ratio is more appropriate to reflect the CD8^+^ TIL relative number, which may provide more actual insight into the functional impact of a CD8^+^ T cell; and (2) the quantitative balance between different subsets of TILs is also revealed by the immune cell ratio, which may be more reliable to indicate the immunologic response status on the tumour microenvironment. Therefore, it was deemed reasonable to use the TIL ratio for predicting prognoses in the present study.

We also found that the intratumoural CD8/CD3 ratio, but not the peritumoural CD8/CD3 ratio, was associated with RFS and OS. This implied a difference in the prognostic significance of the TIL ratio between the intratumoural and peritumoural regions of liver metastases, which might be due to the difference in intensity of the immune response at the two regions. We also observed a significantly lower quantity of CD3^+^, CD8^+^, and Foxp3^+^ T cells in the intratumoural region than in the peritumoural region of liver metastases. In line with our results, the findings of Wang et al. and those from the EORTC 40983 study also showed a lower TIL density in the intratumoural regions of liver metastases than in the peritumoural regions [21,23]. To the best of our knowledge, a high tumour burden in the intratumoural microenvironment might promote immunosuppression [26,27]. In this condition, the relative number of CD8^+^ T cells is important to maintain an immune response in order to combat tumour progression, and thus markedly contributes to tumour recurrence and patient survival.

A large number of studies have shown that Tregs promote tumour growth by inducing host tolerance against tumour antigens by attenuating the T cell-mediated immune response against the tumour cells and enabling them to evade the antitumour immune response [28]. Previous studies have reported that Foxp3 expression in CRC was associated with a high clinical risk score (CRS) and poor overall survival [24,25]. However, our data suggest that the Foxp3/CD3 ratio may not influence patient survival. Similarly, the Naohiro Yoshida et al. study examined primary tumours from 199 stage II–III CRC patients and found that the Foxp3/CD3 ratio did not correlate with metastasis or prognosis. Moreover, Foxp3-positive cell density was found to be correlated with improved prognosis in CRC [29]. These findings might account for the relative abundance of tumour-promoting Tregs. In our study, we found that the number of infiltrating Tregs was not a significant determinant of CLO, presenting a lower tumour burden than the more advanced CRLM, and thus showing relatively low Treg immunogenicity.

α-SMA-labelled CAFs were found to be an abundant stromal immune cell population within the liver oligometastasis microenvironment, and they act as suppressive intermediates in the tumour microenvironment (TME) through the secretion of immunomodulatory factors, subsequently regulating tumour invasion and stimulating metastasis [30,31]. Accumulating clinical data have revealed that the presence of CAFs was associated with poorer survival in multiple cancer types, including CRC [32,33,34]. Our results show that the number of α-SMA-positive cells demonstrates a significant correlation with TILs in the intratumoural regions and the peritumoural regions of liver metastases, which indicates that α-SMA-positive cells might promote the infiltration of T cell subpopulations into the tumour microenvironment. 

To date, there is a growing consensus that CD3^+^ and CD8^+^ TILs should be included in the standard tumour pathological scoring for colorectal cancer [18,35]. Tissue typing of the TME composition may help identify patients with advanced cancer who would benefit most from clinical treatment decision, as previously reported [36]. According to the results of the present study, we hypothesize that calculation of intratumoural CD8/CD3 ratio might offer a reference for the use of immune therapy or postoperative chemotherapy. For instance, a high intratumoural CD8/CD3 ratio facilitated the identification of patients with favourable outcomes, and thus those who might benefit less from postoperative chemotherapy but more from immune therapy. Accordingly, enhanced postoperative chemotherapy should be avoided. These patients can therefore be spared from the associated toxicity, cost, and inconvenience of over-treatment. On the other hand, patients with a low intratumoural CD8/CD3 ratio might benefit from aggressive postoperative chemotherapy and stricter follow-up measures, as they would have poorer prognoses. This hypothesis requires further evaluation in subsequent studies.

Our data must be interpreted in the context of the limitations to the study design. First, the limited number of immune biomarkers detected in liver oligometastasis may have contributed to the underestimation of the CLO microenvironment. Some immune checkpoints, such as PD-L1 and PD-1, are representative of the novel immunotherapeutic strategies that are closely associated with T cell-mediated killing and tumour progression [37,38]. Second, our study failed to show specific spatial distributions of tumour cell-adjacent TIL cells, as high infiltration values of tumour cell-adjacent cytotoxic T cells were significantly correlated with favourable survival [36]. Finally, the present study was conducted with an uncontrolled methodology and included a limited number of patients from a single centre. Therefore, these findings need to be validated in multicentre studies or larger cohort studies.

## 5. Conclusions

This study presents tissue typing results showing the of TIL and CAF composition in the microenvironment of the liver oligometastases from colorectal cancer. Specifically, we demonstrated that the intratumoural CD8/CD3 ratio could provide significant independent prognostic information for CLO patients who are undergoing liver resection, thereby demonstrating the strong potential to serve as a valuable prognostic index.

## Figures and Tables

**Figure 1 cancers-11-01922-f001:**
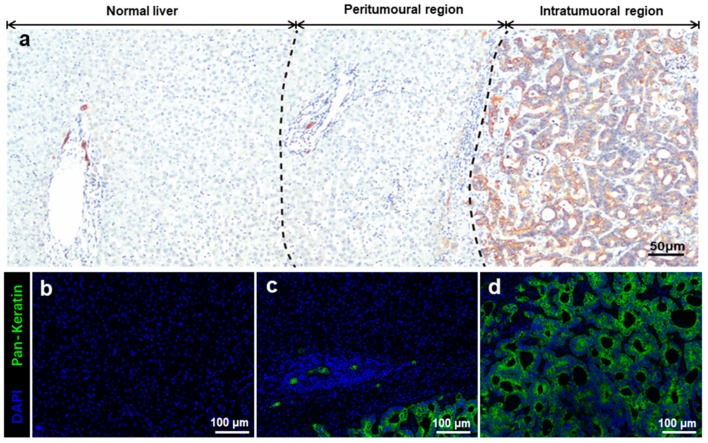
Identification of specific regions of liver oligometastases from colorectal cancer. (**a**) Overview of the immunohistochemical images of different regions of liver oligometastasis. Pan-keratin was highly expressed in the cytoplasm of tumour but not in normal hepatocytes; (**b**) immunofluorescence staining image of normal liver tissue; (**c**) immunofluorescence staining image of the peritumoural region; and (**d**) immunofluorescence staining image of the intratumoural region. Pan-keratin is green (cytoplasm, fluorophore 520), and DAPI is blue. The scale bars of image (**a**) equal 50 μm and images (**b**–**d**) equal 100 μm.

**Figure 2 cancers-11-01922-f002:**
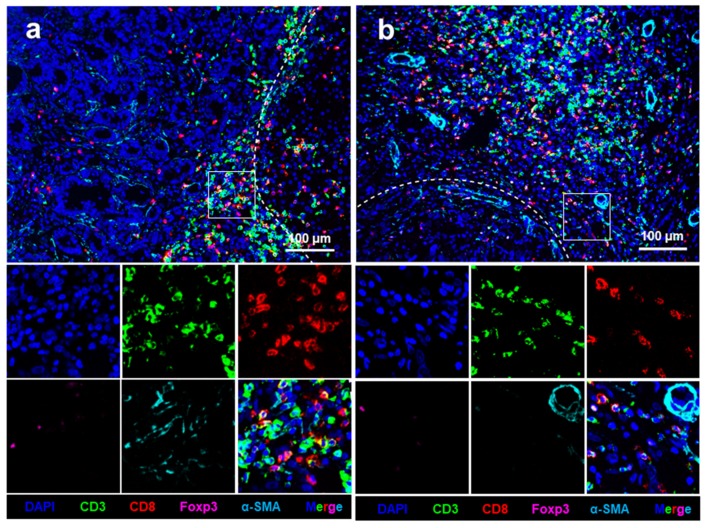
Four-colour immunohistochemical multiplex analysis of liver oligometastasis from colorectal cancer. (**a**) Representative multiplex image of the intratumoural regions of liver metastases; and (**b**) representative multiplex images of the peritumoural regions of liver metastases. Nuclei (DAPI, blue), CD3 (membrane, fluorophore 520, green), CD8 (membrane, fluorophore 690, red), Foxp3 (membrane, fluorophore 620, pink), and α-SMA (cytoplasmic, fluorophore 570, indigo). The white dotted lines represent the borders of liver metastasis and normal liver tissue. All scale bars equal 100 μm.

**Figure 3 cancers-11-01922-f003:**
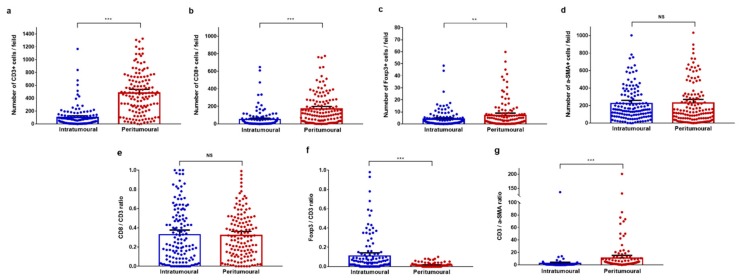
Infiltration of T cell subpopulations and α-SMA-positive cells between intratumoural and peritumoural regions of colorectal liver metastases. (**a**–**f**) Pairwise comparisons of the number of immune cells between intratumoural and peritumoural regions. (**a**) CD3^+^ T cells; (**b**) CD8^+^ T cells; (**c**) Foxp3^+^ T cells; and (**d**) α-SMA^+^ cells. (**e**–**g**) Pairwise comparisons of the tumour-infiltrating lymphocyte ratio between intratumoural and peritumoural regions. (**e**) The CD8/CD3 ratio; (**f**) the Foxp3/CD3 ratio; and (**g**) the CD3/α-SMA ratio. Significance determined by unpaired t-tests. Data are presented as the means with 95% confidence intervals (CIs). ** *p* < 0.01, *** *p* < 0.001; NS, not significant.

**Figure 4 cancers-11-01922-f004:**
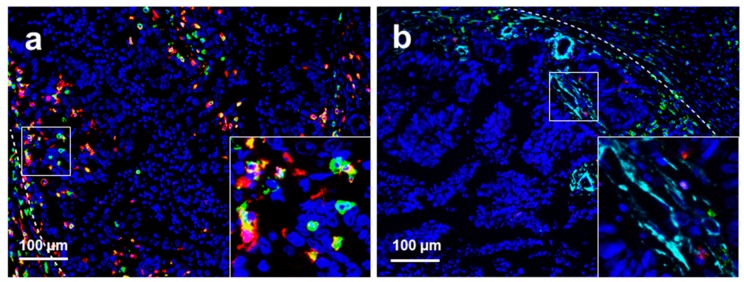
Representative immunohistochemical multiplex images of different CD3/CD8 ratios of T cells in liver metastasis of colorectal cancer. (**a**) High CD8/CD3 ratio in the intratumoural regions of liver metastases; (**b**) low CD8/CD3 ratio in the intratumoural regions of liver metastases; (**c**) high CD8/CD3 ratio in the peritumoural regions of liver metastases; and (**d**) low CD8/CD3 ratio in the peritumoural regions of liver metastases. Nuclei (DAPI, blue), CD3 (membrane, fluorophore 520, green), and CD8 (membrane, fluorophore 690, red). The white dotted lines represent the borders of liver metastases and normal liver tissue. All scale bars equal 100 μm.

**Figure 5 cancers-11-01922-f005:**
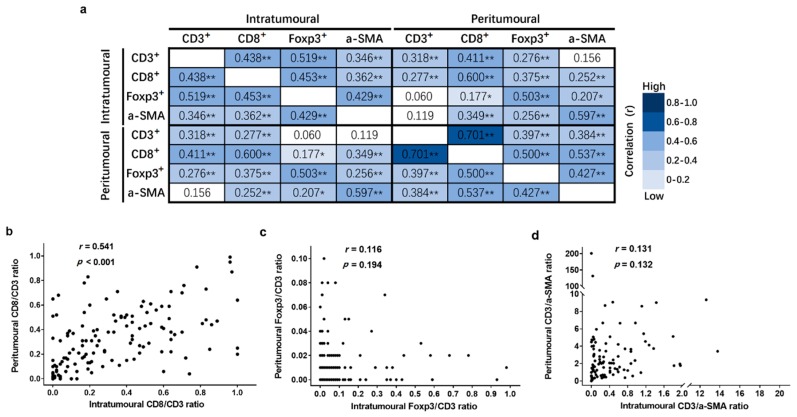
The inter-correlation between different immune cell subtypes in intratumoural and peritumoural regions of liver metastases. (**a**) The inter-correlation between each cell subtype number in intratumoural and peritumoural regions of liver metastases; (**b**) the inter-correlation of the CD8/CD3 ratio in the intratumoural and peritumoural regions of liver metastases; (**c**) the inter-correlation of the Foxp3/CD3 ratio in the intratumoural and peritumoural regions of liver metastases; and (**d**) the inter-correlation of the CD3/α-SMA ratio in the intratumoural and peritumoural regions of liver metastases. Pearson correlation coefficient (r) and significance levels (*p* value) are presented for each correlation. Data are presented as * *p* < 0.05, ** *p* < 0.01.

**Figure 6 cancers-11-01922-f006:**
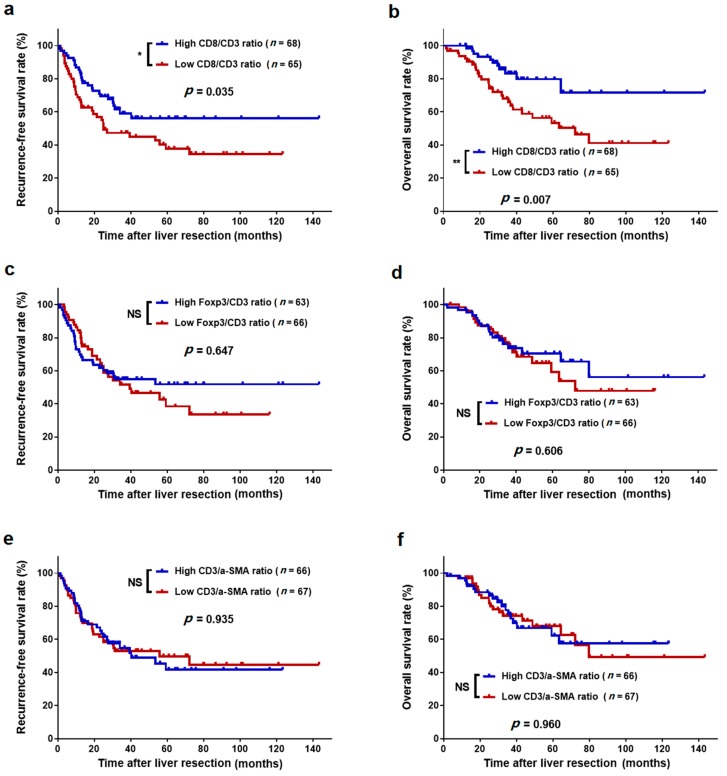
Kaplan–Meier survival curves of colorectal liver oligometastases (CLO) patients after liver metastasectomy stratified by the different levels of tumour-infiltrating lymphocyte ratios in intratumoural regions of liver metastases. (**a**) Recurrence-free survival (RFS) for CD8/CD3 ratio; (**b**) overall survival (OS) for CD8/CD3 ratio; (**c**) RFS for Foxp3/CD3 ratio; (**d**) OS for Foxp3/CD3 ratio; (**e**) RFS for CD3/α-SMA ratio; and (**f**) OS for CD3/α-SMA ratio. * *p* < 0.05, ** *p* < 0.01; NS, not significant.

**Figure 7 cancers-11-01922-f007:**
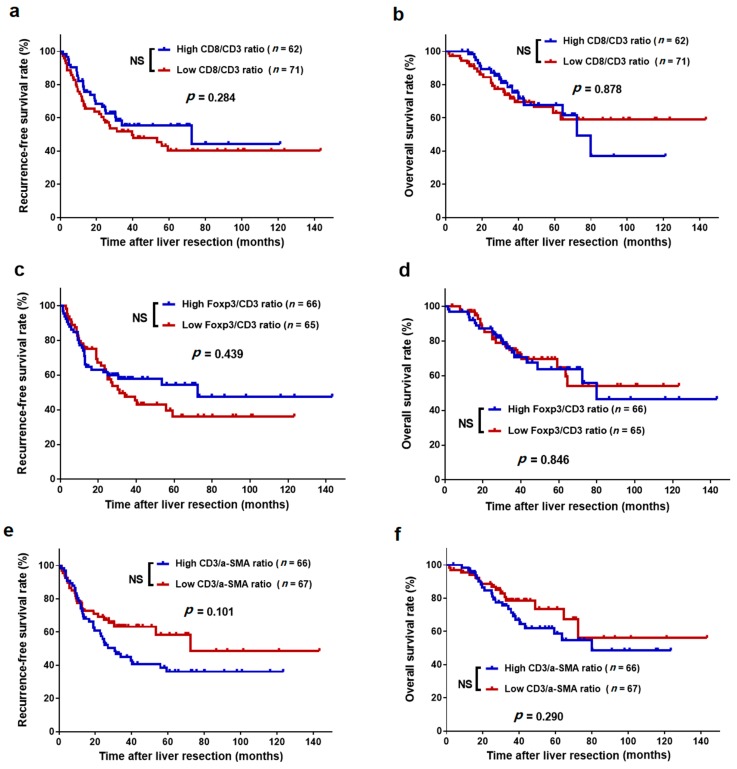
Kaplan–Meier survival curves of CLO patients after liver metastasectomy stratified by the different levels of tumour-infiltrating lymphocyte ratios in the peritumoural regions of liver metastases. (**a**) RFS for CD8/CD3 ratio; (**b**) OS for CD8/CD3 ratio; (**c**) RFS for Foxp3/CD3 ratio; (**d**) OS for Foxp3/CD3 ratio; (**e**) RFS for CD3/α-SMA ratio; and (**f**) OS for CD3/α-SMA ratio. NS, not significant.

**Table 1 cancers-11-01922-t001:** Characteristics of the 133 patients with colorectal liver oligometastasis.

Parameters	Total Patients (*n* = 133, %)
**Patient characteristics**	
Median age (years)	57.8 (25–78)
Age, year	
≤60	82 (61.7)
>60	51 (38.3)
Sex	
Male	80 (60.2)
Female	53 (39.8)
Hepatitis B virus infection	
Negative	114 (85.7)
Positive	19 (14.3)
Primary tumour location	
Right-side colon	34 (25.6)
Left-side colon	50 (37.6)
Rectum	49 (36.8)
Primary tumour differentiation	
Well to moderate	98 (73.7)
Poor	35 (26.3)
T stage ^a^	
1–3	82 (61.7)
4	51 (38.3)
N stage ^a^	
0	44 (33.1)
1–2	89 (66.9)
Timing of metastasis	
Synchronous	86 (64.7)
Metachronous	47 (35.3)
Metastasis diameter (cm)	
Median (range)	2.5 (0.3–12)
≤3	93 (69.9)
>3	40 (31.1)
Median liver resection margin (cm)	1.0 (0–3.5)
MSKCC-CRS	
0	10 (7.5)
1	49 (36.8)
2	66 (49.6)
3	7 (5.3)
4	1 (0.8)
Adjuvant chemotherapy	
Oxaliplatin-based regimen	72 (54.1)
Irinotecan-based regimen	14 (10.5)
5-Fu alone	5 (3.8)
No	42 (31.6)

^a^ Disease stage was classified according to the 2010 American Joint Committee on Cancer/International Union Against Cancer (AJCC/UICC) staging system. Abbreviations: MSKCC-CRS, Memorial Sloan–Kettering Cancer Center clinical risk score.

**Table 2 cancers-11-01922-t002:** The relationship between tumour-infiltrating lymphocyte ratios and clinicopathological features.

	CD8/CD3 Ratio						Foxp3/CD3 Ratio						CD3/α-SMA Ratio					
	Intratumoural (*n* = 133)			Peritumoural (*n* = 133)			Intratumoural (*n* = 129)			Peritumoural (*n* = 131)			Intratumoural (*n* = 133)			Peritumoural (*n* = 133)		
Parameters	High (*n*, %)	Low (*n*, %)	*p* Value	High (*n*, %)	Low (*n*, %)	*p* Value	High (*n*, %)	Low (*n*, %)	*p* Value	High (*n*, %)	Low (*n*, %)	*p* Value	High (*n*, %)	Low (*n*, %)	*p* Value	High (*n*, %)	Low (*n*, %)	*p* Value
Age, year			0.697			0.917			0.828			0.651			0.242			0.804
≤60	42 (61.8)	38 (58.5)		37 (59.7)	43 (60.6)		37 (58.7)	40 (60.6)		28 (62.2)	50 (58.1)		43 (65.2)	37 (55.2)		39 (59.1)	41 (61.2)	
>60	26 (38.2)	27 (41.5)		25 (40.3)	28 (39.4)		26 (41.3)	26 (39.4)		17 (37.8)	36 (41.9)		23 (34.8)	30 (44.8)		27 (40.9)	26 (38.8)	
Sex			0.297			0.936			0.880			0.576			0.337			0.641
Male	39 (57.4)	43 (66.2)		38 (61.3)	44 (62.0)		39 (61.9)	40 (60.6)		26 (57.8)	54 (62.8)		38 (57.6)	44 (65.7)		42 (63.6)	40 (59.7)	
Female	29 (42.6)	22 (33.8)		24 (38.7)	27 (38.0)		24 (38.1)	26 (39.4)		19 (42.2)	32 (37.2)		28 (42.4)	23 (34.3)		24 (36.4)	27 (40.3)	
Hepatitis B Virus infection			0.103			0.670			0.176			0.441			0.229			0.777
Negative	55 (80.9)	59 (90.8)		54 (87.1)	60 (84.5)		51 (81.0)	59 (89.4)		37 (82.2)	75 (87.2)		59 (89.4)	55 (82.1)		56 (84.8)	58 (86.6)	
Positive	13 (19.1)	6 (9.2)		8 (12.9)	11 (15.5)		12 (19.0)	7 (10.6)		8 (17.8)	11 (12.8)		7 (10.6)	12 (17.9)		10 (15.2)	9 (13.4)	
Primary tumour location			0.011			0.035			0.736			0.013			0.056			0.545
Colon	50 (73.5)	34 (52.3)		45 (72.6)	39 (54.9)		40 (63.5)	40 (60.6)		35 (77.8)	48 (55.8)		47 (71.2)	37 (55.2)		40 (60.6)	44 (65.7)	
Rectum	18 (26.5)	31 (47.7)		17 (27.4)	32 (45.1)		23 (36.5)	26 (39.4)		10 (22.2)	38 (44.2)		19 (28.8)	30 (44.8)		26 (39.4)	23 (34.3)	
Primary tumour differentiation			0.044			0.289			0.809			0.098			0.153			0.520
Well to moderate	45 (66.2)	53 (81.5)		43 (69.4)	55 (77.5)		47 (74.6)	48 (72.7)		29 (64.4)	67 (77.9)		45 (68.2)	53 (79.1)		47 (71.2)	51 (76.1)	
Poor	23 (33.8)	12 (18.5)		19 (30.6)	16 (22.5)		16 (25.4)	18 (27.3)		16 (35.6)	19 (22.1)		21 (31.8)	14 (20.9)		19 (28.8)	16 (23.9)	
T stage ^a^			0.492			0.526			0.744			0.342			0.238			0.912
1–3	40 (58.8)	42 (64.6)		40 (64.5)	42 (59.2)		39 (61.9)	39 (59.1)		30 (66.7)	50 (58.1)		44 (66.7)	38 (56.7)		41 (62.1)	41 (61.2)	
4	28 (41.2)	23 (35.4)		22 (35.5)	29 (40.8)		24 (38.1)	27 (40.9)		15 (33.3)	36 (41.9)		22 (33.3)	29 (43.3)		25 (37.9)	26 (38.8)	
N stage ^a^			0.579			0.017			0.253			0.488			0.758			0.758
0	24 (35.3)	20 (30.8)		27 (43.5)	17 (23.9)		17 (27.0)	24 (36.4)		13 (28.9)	30 (34.9)		21 (31.8)	23 (34.3)		21 (31.8)	23 (34.3)	
1–2	44 (64.7)	45 (69.2)		35 (56.5)	54 (76.1)		46 (73.0)	42 (63.6)		32 (71.1)	56 (65.1)		45 (68.2)	44 (65.7)		45 (68.2)	44 (65.7)	
MSKCC-CRS			0.771			0.862			0.057			0.068			0.655			0.426
0–1	31 (45.6)	28 (43.1)		28 (45.2)	31 (43.7)		22 (34.9)	34 (51.5)		15 (33.3)	43 (50.0)		28 (42.4)	31 (46.3)		27 (40.9)	32 (47.8)	
2–4	37 (45.6)	37 (56.9)		34 (54.8)	40 (56.3)		41 (65.1)	32 (48.5)		30 (66.7)	43 (50.0)		38 (57.6)	36 (53.7)		39 (59.1)	35 (52.2)	
Adjuvant chemotherapy			0.188			0.595			0.084			0.574			0.953			0.421
No	25 (36.8)	17 (26.2)		21 (33.9)	21 (29.6)		15 (23.8)	25 (37.9)		13 (28.9)	29 (33.7)		21 (31.8)	21 (31.3)		23 (34.8)	19 (28.4)	
Yes	43 (63.2)	48 (73.8)		41 (66.1)	50 (70.4)		48 (76.2)	41 (62.1)		32 (71.1)	57 (66.3)		45 (68.2)	46 (68.7)		43 (65.2)	48 (71.6)	

^a^ Disease stage was classified according to the 2010 American Joint Committee on Cancer/International Union Against Cancer (AJCC/UICC) staging system. Abbreviations: MSKCC-CRS, Memorial Sloan–Kettering Cancer Center clinical risk score.

**Table 3 cancers-11-01922-t003:** Univariate and multivariate analyses for recurrence-free survival and overall survival in patients undergoing liver metastasectomy.

Parameters	RFS				OS			
	Univariate		Multivariate		Univariate		Multivariate	
	HR (95% CI)	*p* Value	HR (95% CI)	*p* Value	HR (95% CI)	*p* Value	HR (95% CI)	*p* Value
Age (>60 years vs. ≤60 years)	0.968 (0.582–1.610)	0.901			1.264 (0.664–2.408)	0.475		
Sex (male vs. female)	1.076 (0.647–1.789)	0.778			0.888 (0.466–1.693)	0.718		
Primary tumour location (rectum vs. colon)	1.302 (0.788–2.152)	0.303			1.361 (0.718–2.581)	0.345		
Primary tumour differentiation (poor vs. well to moderate)	1.327 (0.773–2.277)	0.305			0.835 (0.382–1.824)	0.652		
T stage (4 vs. 1–3)	1.341 (0.813–2.212)	0.250			1.524 (0.802–2.896)	0.199		
N stage (1–2 vs. 0)	1.844 (1.044–3.256)	0.035	1.806 (1.018–3.202)	0.043	1.922 (0.909–4.067)	0.087	2.041(0.960–4.338)	0.064
Timing of metastasis (synchronous vs. metachronous)	1.220 (0.718–2.074)	0.463			1.245 (0.628–2.470)	0.531		
Metastases diameter (>3 cm vs. ≤3 cm)	1.576 (0.943–2.634)	0.083	1.533 (0.915–2.568)	0.105	1.870 (0.986–3.547)	0.055	1.738 (0.915–3.300)	0.091
Preoperative CEA (>50 ng/mL vs. ≤50 ng/mL)	1.477 (0.810–2.694)	0.204			1.796 (0.878–3.674)	0.109		
Adjuvant chemotherapy (yes vs. no)	1.794 (0.973–3.307)	0.061	1.666 (0.900–3.081)	0.104	1.567 (0.688–3.570)	0.285		
Intratumoural CD8/CD3 ratio (high vs. low)	0.585 (0.353–0.968)	0.037	0.593 (0.357–0.985)	0.043	0.392 (0.193–0.793)	0.009	0.391 (0.193–0.794)	0.009

Abbreviations: CEA, carcinoembryonic antigen; RFS, recurrence-free survival; OS, overall survival; HR, hazard ratios; CI, confidence interval. Disease stage was classified according to the 2010 American Joint Committee on Cancer/International Union Against Cancer (AJCC/UICC) staging system.

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
