# Peer review of "Immune Cell Infiltration in the Microenvironment of Liver Oligometastasis from Colorectal Cancer: Intratumoural CD8/CD3 Ratio Is a Valuable Prognostic Index for Patients Undergoing Liver Metastasectomy"

_cancers, 2019, doi:10.3390/cancers11121922_

Round 1

Reviewer 1 Report

In this study, Jianhong Peng et al. have analyzed the prognostic significance of immune cell or myofibroblast (CD3, CD8, FOXP3, alpha-SMA) ratios colorectal cancer liver oligometastasis tumor microenvironment in a consecutive single-institutional cohort of 133 patients, using multiplex immunofluorescence.

I think the topic of the study is important, considering increasing success in the treatment of colorectal cancer liver metastases with surgical approaches, as well as increasing interest in tumor immunology. The authors utilize novel, practical method for the simultaneous detection of multiple protein markers in one tissue section. However, a more accurate description of this method and its validation might be useful to ensure the reliability of the results. The authors state they have taken ethical considerations into account in the study design (Approval from Institutional Ethics Committee).

My specific comments to improve the manuscript are listed below:

Major comments:

The authors have used multiplex immunofluorescence (mIF) assay to detect CD3, CD8, FOXP3, alpha-SMA, and cytokeratin in formalin-fixed paraffin-embedded liver metastasis tissue. During the past few years, more and more laboratories have adopted different methods of mIF. Accurate results require proper validation and calibration of this sensitive method. Therefore, I would recommend adding data of quality control and validation of the assay by, for example, using similar approaches (comparison to standard IHC in multiple samples, evaluation of batch effects, etc.) as in (Parra ER, et al. Sci Rep. 2017;7:13380).

The authors have assessed the relationships between several types of tumor-infiltrating immune cells/fibroblasts and a high number of other variables. I think multiple hypothesis testing represents one of the most important limitations of this study, which should be highlighted in the discussion section. There is a divergence of opinion, ranging from a strict view of using a Bonferroni or other adjustment for all multiple comparisons, to the view articulated by Rothman and others, who deny the existence of a multiple comparison problem. Some authors have recommended more strict p-value limits for claims of new discoveries than 0.05, such as 0.005 (Benjamin DJ, et al. Nat Hum Behav. 2018;2:6–10). If the authors do not decide to apply adjustments to the level of statistical significance due to multiple hypothesis testing, I would still recommend interpreting the findings with p-value near 0.05 with caution.

Minor comments:

Accurate and complete reporting of a tumor prognostic marker study facilitates the assessment of study quality, aids understanding of the relevance of the study conclusions, and is important for the reproducibility of the results. For these studies, REMARK guidelines are particularly useful. The authors have covered most items of this checklist in their manuscript. However, I would still recommend reviewing this checklist in detail and adding any missing information such as: “Specify all statistical methods, including details of any variable selection procedures and other model-building issues, how model assumptions were verified, and how missing data were handled. (REMARK 10)” How did the authors verify the assumptions of Cox proportional hazards regression?

The mIF staining protocol needs to be more accurately described. The current protocol does not enable the replication of this study. Moreover, catalog numbers (or preferably, RRIDs) should be provided for each reagent.

The terms “intratumoral and peritumoral metastases” on line 32 is a bit confusing.

I would recommend stating “potentially curative” instead of “curative” on line 86.

In section 2.2., there is some discrepancy, as the authors mention five markers in some sentences but four markers in some. I would recommend addressing this. Was cytokeratin immunofluorescence conducted on all the tumors as a separate slide?

On line 132-133, the authors state, the cells were phenotyped into five classes: CD3+ T cells, CD8+ T cells, 132 Foxp3+ T cells, α-SMA+ fibroblasts, and CD3- CD8- Foxp3- α-SMA- cells. However, these classes are overlapping, as CD3 is usually expressed in CD8+ and FOXP3+ cells. Please further specify this.

Starting from line 180, the authors list “mean values of absolute numbers” for each cell type. Have these numbers been standardized according to the examined area? In that case, they are not absolute numbers but densities, and the standardization factor (area) should be provided? If they have not been standardized, the numbers may not be comparable.

The authors state alpha-SMA was used to examine myofibroblasts. However, the images the authors provide also show alpha-SMA-positivity in blood vessel smooth muscle. Did the authors exclude these structures from the analysis? It should be mentioned in the text that alpha-SMA is not specific for myofibroblasts but is also expressed by smooth muscle in general, including many blood vessels.

I think non-parametric paired test (Wilcoxon signed-rank test) would be more appropriate for Fig. 3 than unpaired t-test, considering the skewed distribution of the variables, and comparison between different regions of the same tumors.

I would recommend analyzing the prognostic value of immune cell densities in addition to cell ratios. If this a secondary analysis not originally planned, it could be presented as supplementary material. However, it would be important to know, whether ratios provide additional value over densities.

The cut-off point for FOXP3/CD3 ratio does not seem to be median according to the numbers of patients in high and low groups.

On line 355, the authors state: “Moreover, Foxp3 expression was found to be correlated with improved prognosis in CRC.” FOXP3 positive cell density instead of expression?

On lines 373-379, the authors hypothesize the results might be used to guide treatment. I would recommend further highlighting that these are just hypotheses, which require evaluation in subsequent studies.

Author Response

Please find the response in the attachment.

Reviewer 2 Report

In the manuscript entitled, Immune cell infiltration in liver oligometastasis 2 microenvironment from colorectal cancer: 3 intratumoural CD8/CD3 ratio represents a valuable 4 prognostic index for patients undergoing liver metastasectomy  Jianhong Peng and colleagues try to find a correlation between the tumor immune microenvironment and patients with colorectal cancer undergoing liver metastasectomy.   The authors employed a five-colour immunohistochemical multiplex technique to simultaneous detect the expression of the following immune markers CD3+, CD8+, and Foxp3+ T cells and α-SMA fibroblasts in the liver oligometastasis.

The topic of this study is original, in the light of growing interest and encouraging results of immunotherapy for the treatment of some cancer subtypes.  Understanding how and if immune cells populations play a role in the metastatic process remains challenging. This is particularly true if we consider patients undergoing liver metastasis (oligometastasis or widespread) for whom clinical options are currently insufficient.

The study is clearly designed, however some limitations exist regarding to the methodological approach and the results.

Major points

1)    The first limitation of the present study is the lack of the molecular classification of the tumors. It is well-known that primary CRCs having mismatch repair (MMR) defects (around 5%) and high CD8+ intatumoral infiltration respond to immunotherapy. Unfortunately, the large majority of CRCs are MMR proficient, recognized as “immune-ignorant or with highly inflamed” tend to metastasize in distant organs. In the case of liver oligometastasis, what type of molecular subtypes are those indicated as having high and low CD8+/CD3+ ratio?. Please indicate through the comparison with primary tumors  if CD8+ cells intra and peri-tumoral  present a comparable recurrence of immune cells.

2)    To address the above questions may provide important  indications on how and if the infiltration of cytotoxic T cells change  through metastatic evolution.  Moreover, this may be relevant to understand if TIL infiltration have comparable relevance both in the primary CRCs and oligometastasis. Can authors include and comments these aspects with available biological material?

3)    Another vulnerable point of the present study is the absence of  markers representative of additional immune populations. For example, the authors have totally ignored some markers of the innate immune response (macrophages, neutrophils) for which a significance role in promoting metastatic progression has been reported in several studies.

4)    Looking the data provided in Figure 3, we don’t ‘know if high CD8+/CD3+ ratio play an effective role in empowering immune system and consequently patients’ prognosis. Perhaps, the analysis of additional immune markers “for example against exhaustive T cells” or NKs may provide an initial indication and partially sustain authors ‘data.

5)     Another concerns regards immune positivity for FOXP3 cells. Some studies have shown that FOXP3 may be also expressed in CRC cells from primary resected tumors. To exclude a such event Pan-Keratin may be used in double immunofluorescence staining.  Notably, the authors should also consider that functionally distinct subpopulations of tumor-infiltrating FOXP3(+) T cells can be found in CRCs (Saito T et al, 2016). Moreover, itCD8(+) T/itFOXP3(+) cell ratio it has also been positively correlated with disease-free survival (0.023) and overall survival (P = 0.010) in CRC (Suzuki H  et, 2010).

6)    In addition, the relationship between TILs ratio and clinicopathologic features is confusing. Please discuss in more detail why  CD3/α-SMA and Foxp3/CD3 ratio, in contrast to CD3+/CD8+ ratio, are correlated with a larger number of clinicopthalogical parameters. Looking at the table 2, Foxp3/CD3 appear to play  a role in MSKCC-CRS classification and adjuvant therapy.

Mino points

1)    Please revise the table 2 highlighting into table what values should be considered.

2)    The values reported in the table in Figure 5a should be revised and corrected according to the scale of values reported on the right.

3)    Please include Pan-190 Keratin in example multiplex analyses.

4)    Please do not overstate the conclusion through the manuscript. Actually, the manuscript does not improve our understanding on the nature of the immunity in metastasis.

5)    The authors should greatly revise the literature provided.

Author Response

(The authors gave the same response as above.)

Reviewer 3 Report

The authors present a multiplexed immunohistochemical analysis of liver metastasis in an attempt to define the tumour microenvironment and identify factors predictive of outcome. 

This is topical and the use of a multiplexed approach is interesting as there is increasing focus on the understanding of the spatial arrangement of the TME. However, the technology utilised limits the analysis to a relatively small number of markers when compared to other technologies (e.g. AKOYA CODEX or GE Cell dive systems) and so that novelty of this work in comparison to other published multiplexed analysis of the colorectal cancer TME (see. Schurch et al Cell 2019) is limited. Equally, the markers utilised are broad (e.g. SMA defines a range of fiborblast subtypes, CD8 and CD3 are relatively non-specific) and so the information gained from this is relatively limited from a biological perspective. 

The finding that ratios of various immune subsets within the TME predict outcome is not particularly novel but rather is supportive of prior literature. 

Points requiring correction:

The information presented in Table 2 is overly complex and should be dramatically simplified.

Why was there need to perform so many comparisons. Further, if so many comparisons are to be performed then the authors need to take account of the false discovery rate which has not been done.

I would suggest dramatically simplifying the table or removing it and its analysis altogether as I do not feel that drawing significance between clinical parameters and the IHC findings is relevant unless the authors can come up with valid biological reasons for doing so.

There was no mention of blinding of the person analysing the staining results to the clinical outcome which would have been important to avoid bias. 

Why was further analysis of the TME not done (e.g. spatial relationship between different cell types etc..), which may be relevant from a biological perspective?

Author Response

(The authors gave the same response as above.)

Round 2

Reviewer 1 Report

I think the authors have appropriately responded to the issues brought up in the review.

Reviewer 2 Report

The authors have replied to many comments raised in the previous assessment.

Although some questions raised remain still open, I can consider the manuscript  satisfying improved in several points.